# Convolutional Neural Network Algorithms for Semantic Segmentation of Volcanic Ash Plumes Using Visible Camera Imagery



**José Francisco Guerrero Tello [1],\***[ID]**, Mauro Coltelli [1]**[ID]**, Maria Marsella [2]**[ID]**, Angela Celauro [2]**[ID]
**and José Antonio Palenzuela Baena [2]**

[1] Istituto Nazionale di Geofisica e Vulcanologia, Osservatorio Etneo, Piazza Roma 2, 95125 Catania, Italy
[2] Department of Civil, Building and Environmental Engineering, Sapienza University of Rome, Via Eudossiana 18, 00184 Roma, Italy
**\*** Correspondence: francisco.guerrero@ingv.it

**Abstract:** In the last decade, video surveillance cameras have experienced a great technological advance, making capturing and processing of digital images and videos more reliable in many fields of application. Hence, video-camera-based systems appear as one of the techniques most widely used in the world for monitoring volcanoes, providing a low cost and handy tool in emergency phases, although the processing of large data volumes from continuous acquisition still represents a challenge. To make these systems more effective in cases of emergency, each pixel of the acquired images must be assigned to class labels to categorise them and to locate and segment the observable eruptive activity. This paper is focused on the detection and segmentation of volcanic ash plumes using convolutional neural networks. Two well-established architectures, the segNet and the U-Net, have been used for the processing of in situ images to validate their usability in the field of volcanology. The dataset fed into the two CNN models was acquired from in situ visible video cameras from a ground-based network (Etna_NETVIS) located on Mount Etna (Italy) during the eruptive episode of 24th December 2018, when 560 images were captured from three different stations: CATANIA-CUAD, BRONTE, and Mt. CAGLIATO. In the preprocessing phase, data labelling for computer vision was used, adding one meaningful and informative label to provide eruptive context and the appropriate input for the training of the machine-learning neural network. Methods presented in this work offer a generalised toolset for volcano monitoring to detect, segment, and track ash plume emissions. The automatic detection of plumes helps to significantly reduce the storage of useless data, starting to register and save eruptive events at the time of unrest when a volcano leaves the rest status, and the semantic segmentation allows volcanic plumes to be tracked automatically and allows geometric parameters to be calculated.

**Keywords:** ANN; automatic classification; risk mitigation; machine learning

## 1. Introduction

Volcano monitoring is composed of a set of techniques that enable the measurement of different parameters (geochemical, seismic, thermal, deformational, etc.) [1]. Keeping these parameters under surveillance is essential for risk mitigation and guarantees security to the population. These parameters allow us to know the state of internal and external activity of a volcano and to know if there are changes in the behaviour of the volcano that can lead to an eruption or to understand if there are changes during an eruptive event. Although seismic and geodetic instruments permit quasi-real-time monitoring, video cameras are also currently a standard and necessary tool for effective volcano observation [2,3].

Explosive volcanic eruptions eject a big quantity of pyroclastic products into the atmosphere. In these events, continuous surveillance is mandatory to avoid significant damage in rural and metropolitan areas [4] that may disrupt the surface and air traffic [5],

and even may cause negative impacts on human health [6]. In 1985, the eruption of "Nevado del Ruiz" volcano in Colombia ejected more than 35 tons of pyroclastic flow that reached 30 km in height. This eruption melted the ice and created four lahars that descended through the slopes of the volcano and destroyed a whole town called "Armero" located 50 km from the volcano, with a loss of 24.800 lives [7]. To counteract further disasters, it is fundamental to create new methodologies and instruments based on innovation for risk mitigation. Video cameras have proven suitable for tracking those pyroclastic products in many volcanoes in the world, whether with visible (0.4–0.7 μm) or near-infrared (~1 μm) wavelength. Both sensors are suitable to collect and analyse information at a long distance.

Video cameras installed on volcanoes often experience limited performance in relation to crisis episodes. They are programmed to capture images in a specific time range (i.e., one capture per minute, one capture every two minutes, etc.); those settings lead to the storage of unnecessary data that need to be deleted manually by an operator with time-consuming tasks. On the other hand, video cameras do not have an internal software to deeply analyse images in real time. This work is carried out after downloading by applying different computer vision techniques to calibrate the sensor [8] and extract relevant information by edge-detection algorithms and GIS-based methods, such as contours detections and statistics classification, such as PCA [9]. All these kinds of postprocessing procedures involve semi-automatics and time-consuming tasks.

These limitations can be faced through machine-learning techniques for computing vision. In the last decade, technological innovation has increased dramatically in the world of artificial intelligence (AI) and machine learning (ML) in parallel to video cameras [10]. The convolutional neural networks (CNN) became popular because they outperformed any other network architecture on computer vision [11]. Specifically, the architecture U-Net is nowadays being routinely and successfully used in image processing, reaching an accuracy similar to or even higher than other existing ANN, for example, of the FCN type [12–14], providing multiple applications where pattern recognition and feature extraction play an essential role. CNNs have been applied to find solutions to mitigate risk in different environmental fields, such as for the detection and segmentation of smoke and forest fires [15,16], flood detection [17], and to find solutions regarding global warming, for example, through monitoring of the ice of the poles [18,19]. CNNs have been applied in several studies in the field of volcanology for earthquake detection and classification [20,21], for the classification of volcanic ash particles [22], and to validate their capability for real-time monitoring of the persistent explosive activity of Stromboli volcano [23], for video data characterisation [2], detection of volcanic unrest [24], and volcanic eruption detection using satellite images [25–27]. Thus, the importance of applying architectures based on CNN could be an alternative to improve the results obtained in the different scientific works performed till now.

This research aims to create algorithms that help solve computer vision problems based on deep learning for the detection and segmentation of the volcanic plume, providing an effective tool for emergency management to risk management practitioners. The concept of this tool focuses on a neural network which is fed with data from the 24th to 27th December 2018 eruptive event. The eruption that began at noon was preceded by 130 earthquake tremors, the two strongest of which measured 4.0 and 3.9 on the Richter scale. From this eruptive event, 560 images were collected and then preprocessed and split into 80% training and 20% validation. The training dataset was used in the training of two very consolidated models: the SegNet Deep Convolutional Encoder-Decoder and U-net architectures. In this groundwork phase, more consolidated models were sought to have a large comparative pool and to substantiate their use in the volcanological field. As a result, a trained model is generated to automatically detect the beginning of an eruptive activity and tracking the entire eruptive episode. Automatic detection of the volcanic plume supports volcanic monitoring to store useful information enabling real-time tracking of the plume and the extraction of concerning geometric parameters. By developing a comprehensive and reliable approach, it is possible to extend it to many other explosive volcanoes. The current

results encourage a broader research objective that will be oriented towards the creation of more advanced neural networks [2], deepening the real-time monitoring for observing precursors, such as change in degassing state.

## 2. Geological Settings

Mt. Etna is a basaltic volcano located in Sicily in the middle of Gela-Catania foredeep, at the front of the Hyblean Foreland [28] (Figure 1). This volcano is one of the most active in the world with its nearly continuous eruptions and lava flow emissions and, with its dimensions, it represents a major potential risk to the community inhabiting its surroundings.

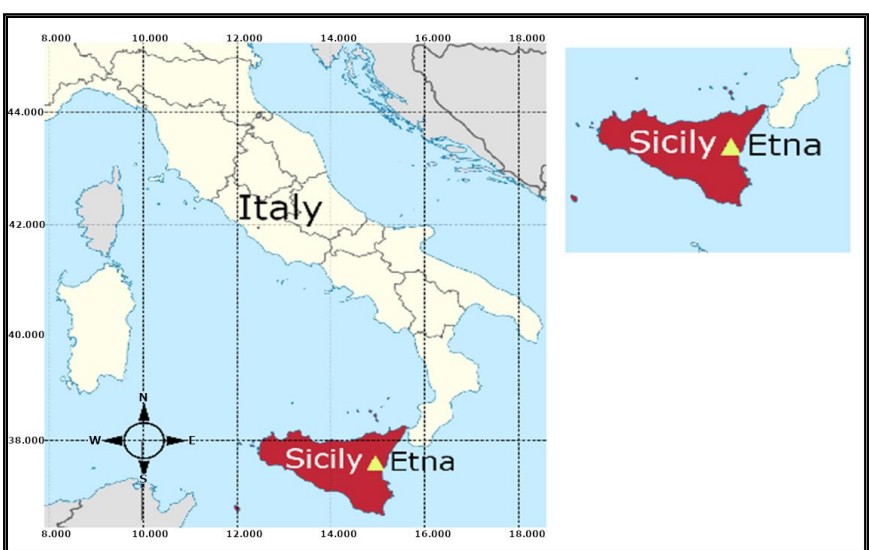

**Figure 1.** Location of Etna volcano.

The geological map, updated in 2011 [29] at the scale of 1:50,000, is a dataset of the Etna eruptions that occurred throughout its history (Figure 2, from [29], with modifications). This information is fundamental for land management and emergency planning.

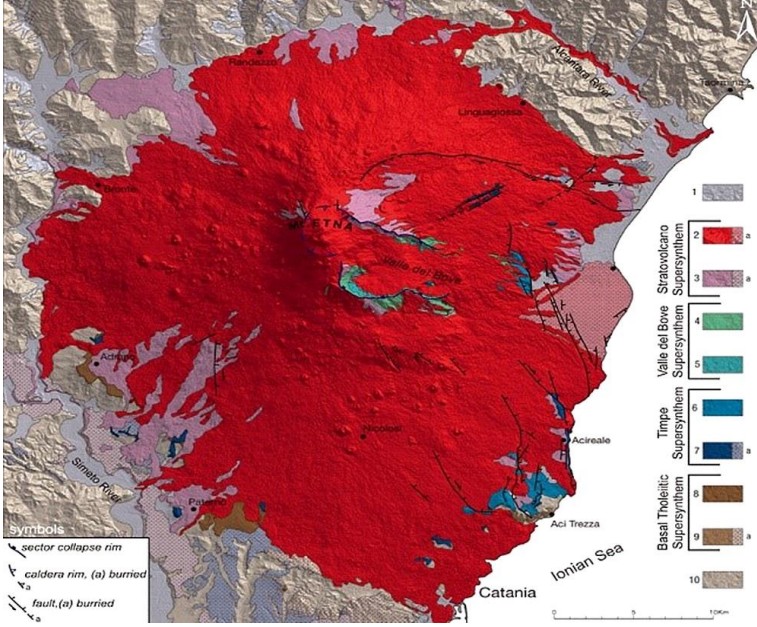

**Figure 2.** Geological map of Mt. Etna.

### 3. Etna_NETVIS Network

Mt. Etna has become one of the better monitored volcanoes in the world by using several instrumental networks. One of them is the permanent terrestrial Network of Thermal and Visible Sensors of Mount Etna, which comprises thermal and visible cameras located at different sites on the southern and eastern flanks of Etna. The network, initially composed of CANON VC-C4R visible (V) and FLIR A40 Thermal (T) cameras installed in Etna Cuad (ECV), Etna Milo (EMV), Etna Montagnola (EMOV and EMOT), and Etna Nicolosi (ENV and ENT), has been recently upgraded (since 2011) by adding high-resolution (H) sensors (VIVOTEK IP8172 and FLIR A320) at the Etna Mt. Cagliato (EMCT and EMCH), Etna Montagnola (EMOH), and Etna Bronte (EBVH) sites [3]. Visible spectrum video cameras used in this work and examples of field of view (FOV), Bronte, Catania, and Mt. Cagliato are shown in Figure 3. These surveillance cameras do not allow 3D model extraction due to poor overlap, unfavourable baseline, and low image resolution. Despite this, simulation of the camera network geometry and sensor configuration have been carried out in a previous project (MEDSUV project [3]) and will be adopted as a reference for future implementation of the Etna Network.

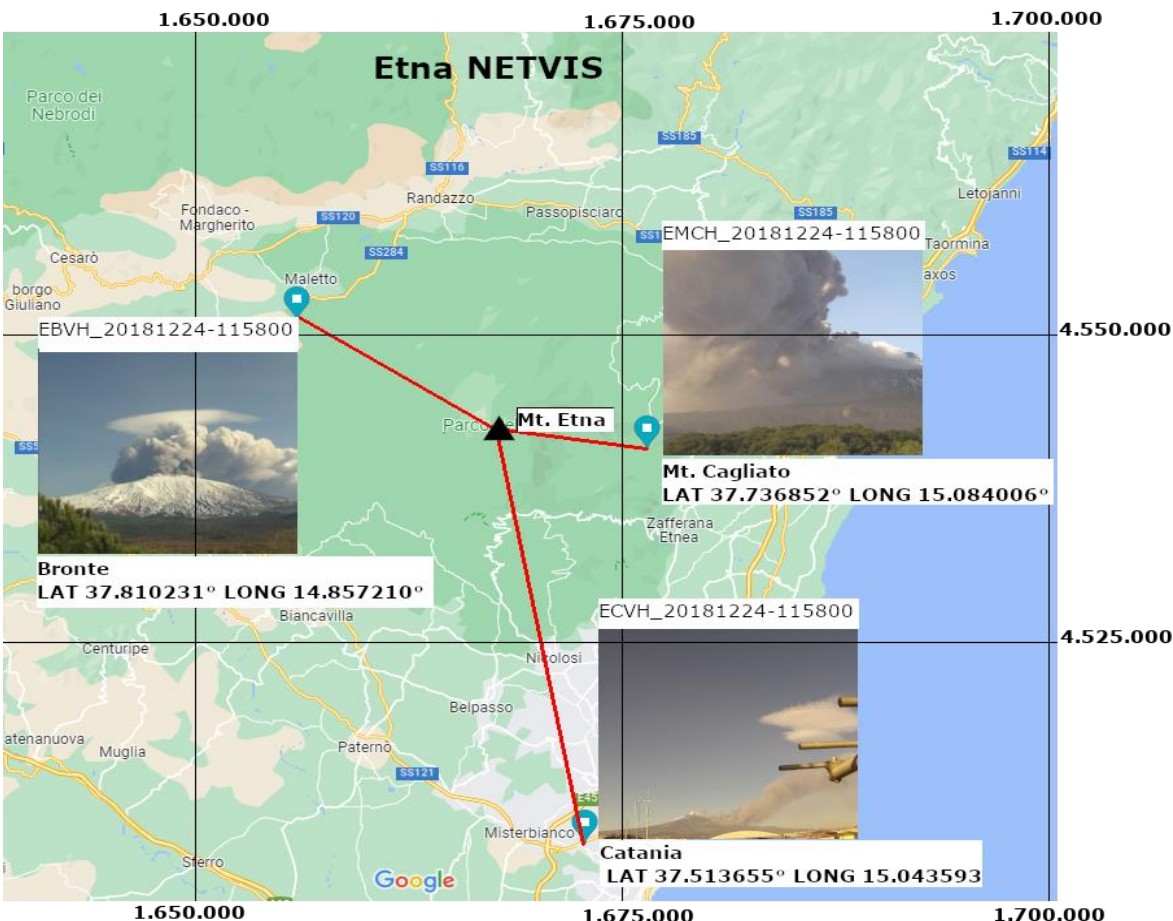

**Figure 3.** Etna_Netvis surveillance network.

The technical specifications of Etna_NETVIS network cameras used in this work, such as pixel resolution, linear distance to the vent, and horizontal and vertical field of view (HFOV and VFOV), are described in Table 1.

**Table 1.** Characteristics of the ETNA NETVIS cameras.

| | ETNA NETVIS | | | | |
|---|---|---|---|---|---|
| Station Name | Resolution Pixel | Distance to the Vent | Image Captured per Minute | Model | Angular FOV (deg) |
| BRONTE | $760 \times 1040$ | 13.78 km | 1 | VIVOTEK | 33_~93_ (horizontal), 24_~68_ (vertical) |
| CATANIA | $2560 \times 1920$ | 27 km | 1 | | |
| MONTE CAGLIATO | $2560 \times 1920$ | 8 km | 2 | VIVOTEK | 33_~93_ (horizontal), 24_~68_ (vertical) |

## 4. Materials and Methods

### 4.1. Materials: Data Preparation

The paradigm used for this work was a supervised learning based on a set of samples consisting of a pair of data; input variables (x) and output labelled variables (y). Data labelling is the crucial part of the data preprocessing in the workflow to build a neural network model, which requires large volumes of high-quality training data. The processes for creating label data are expensive, complicated, and time-consuming. Many open-source libraries, such as MNIST by Keras, offer a full dataset ready to use, but it covers neither all types of objects nor labelled data for volcanic ash plume shapes. Thus, the 560 images collected were manually labelled using an open-source image editor "GIMP" to delineate the boundaries of volcanic plums and generate the ground truth mask (Figure 4). The samples were split into two sets: training and validation in a proportion of 80% and 20%, respectively. As this research deals with a binary classification problem, the neural network is contextualised within volcanic plume shapes by assigning pixel level. Thus, pixels that are inside the ash column contour are assigned values of 255 or, otherwise, 0. Inputs with large integer values could collapse the bias value or slow down the learning process, so, to avoid this effect, pixels were normalised between 0 and 1 by applying Equation (1):

$$x' = \frac{(x - x_{min})}{(x_{max} - x_{min})} \tag{1}$$

where $x$ is the pixel to normalize, $x_{min}$ is the minimum value of pixels of the image, and $x_{max}$ is the maximum value pixel of the image. To keep size consistency across the dataset while reducing memory consumption, images were resized to (768px $\times$ 768px) by applying bilinear interpolation.

Finally, to improve the robustness of the inputs, the training data were augmented through a technique called "data augmentation". It was applied with the Keras library "ImageDataGenerator" class that artificially expands the size of the dataset, creating some perturbating in our images as horizontal flips, zoom, random noise, and rotations (Figure 5). Data augmentation avoids overfitting in the training stage.

### 4.2. Methods: ANN and UNET

The perceptron, core concept of deep learning and convolutional neural network introduced by Rosenblatt [30], in brief, consists of a single-layer neural network whose base algorithms are the threshold function and the gradient descent [31]. The latter method is the most popular algorithm that performs parametrisation and optimisation of the parameters in the artificial neural network (ANN), by means of labelled samples and process iterations for the prediction of accurate outputs [31].

The optimisation minimises the loss function (or cost function), represented by the cross-entropy as a measure of the difference between the actual and predicted classes. Finally, the learning rate is an important parameter, used in the following sections to control the time of the algorithm and the network parameter training at every iteration, which is crucial to reach the expected results of the refined model. These parameters are here briefly introduced, leaving the theoretical digression to dedicated sources [30,31].

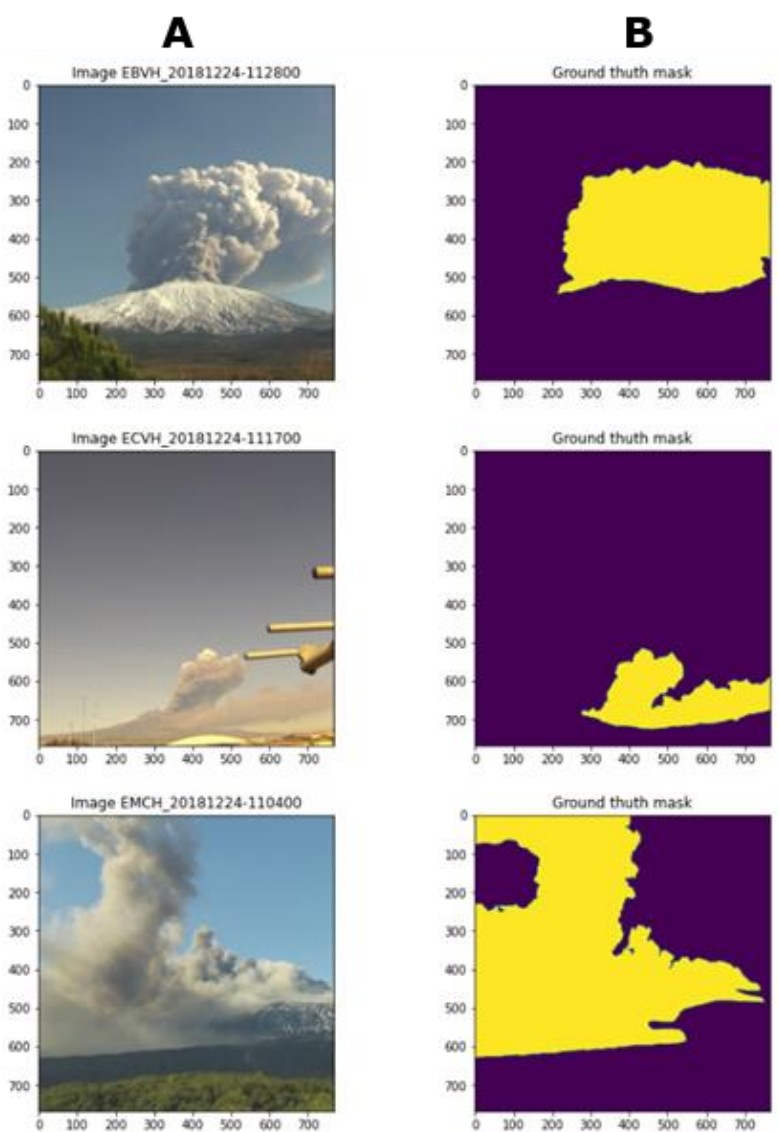

**Figure 4.** Examples of variable pairs (in (**A**) the real images are shown and (**B**) represents the ground truth mask).

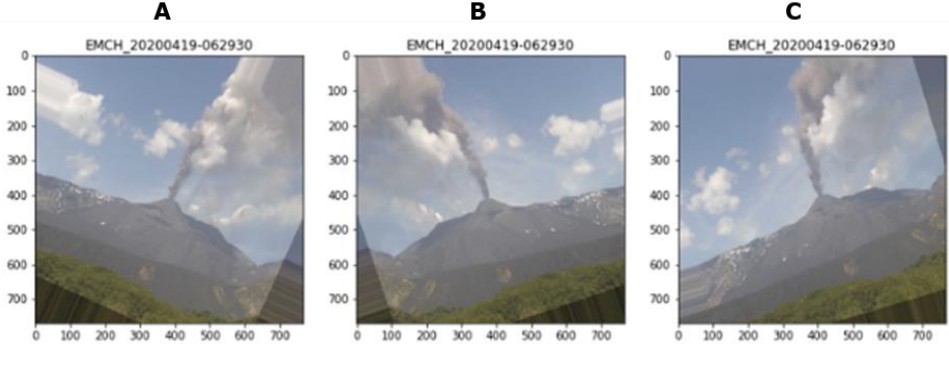

**Figure 5.** Example of data augmentation with vertical and horizontal flips ((**A**) is a vertical right flipped image of 60 inclination degrees, (**B**) is a horizontal and vertical flipped and (**C**) is a horizontal and vertical flipped with distortion).

Convolutional Neural Network Architectures

Segmentation is a fundamental task for image analysis. Semantic segmentation describes the process of associating each pixel in an image with a class label. Segmenting images of volcanic plumes is a complicated task, different from segmenting other objects, such as people, cars, roads, buildings, and other entities that are well differentiated from their background. Those types of objects are considered homogeneous and regular in form and radiometry, but a volcanic plume can have very different physical properties [32], such as shapes, colour, and density. In deep learning, CNN appears as a class of ANN based on the shared-weight architecture of the convolution kernels [11] and proved very efficient for pattern recognition, feature extraction for applications in computer vision analysis and image recognition [33], classification [34], and segmentation [35]. This is useful to solve problems as faced in this paper. Thus, this paper presents developed models based on specific CNN architectures.

Different algorithms were implemented to develop a tool able to segment a volcanic ash plume from in situ images, creating two models based on architectures of Seg-Net [36] and U-Net [37]. Those trained models were carried out using Tensorflow GPU version 2.12 [38], Python 3.6 language, and Keras 2.9 [39], all of these based on open-source libraries and built on Tensorflow framework. Keras appears here as the core language for ANN programming, as it contains numerous implementations of commonly used neural network building blocks, such as layers, activation functions, optimizers, metrics, and tools, to preprocess images.

The U-net (Figure 6) is a CNN architecture for the segmentation of images, developed by Olaf Ronneberger et al. [37] and used for medical scope, but now applied in several other fields [40–43]. It is built upon the symmetric fully convolutional network and is made up of two parts. The down-sampling network (encoder) reduces dimensionality of the features while losing spatial information; instead, the up-sampling network (decoder) enables the up-sampling of an input feature map to a desired output feature map using some learnable parameters based on transposed convolutions. Thus, it is an end-to-end fully convolutional network (FCN) that makes it possible to accept images of any size.

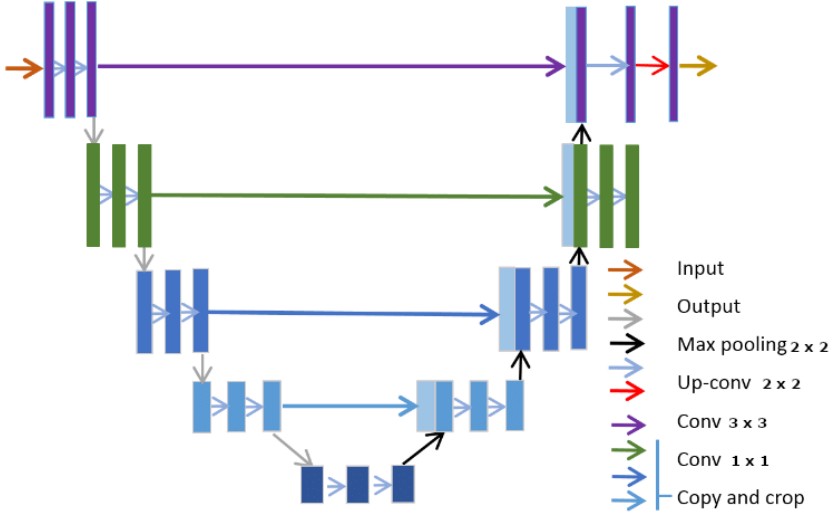

**Figure 6.** U-net architecture.

On the other hand, the SegNet architecture [36] FCN is based on decoupled encoder–decoder, where the encoder network is based on convolutional layers, while the decoder is based on up-samples. The architecture of this model is shown in Figure 7. It is a symmetric network where each layer of encoder has a corresponding layer in the decoder.

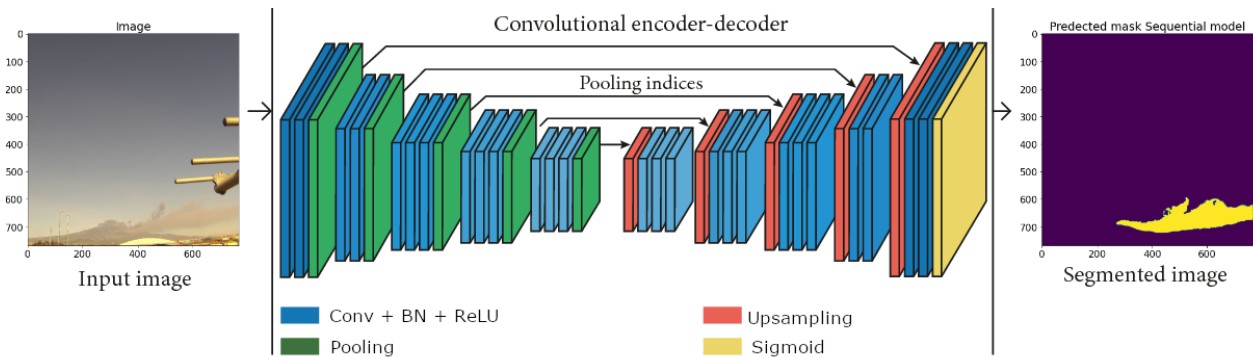

**Figure 7.** SegNet architecture.

Loss functions are used to optimize the model during training stage, aiming at minimising the loss function (error). The lower the value of loss function, the better the model. Cross-entropy loss is the most important loss function to face classification problems. The problem tackled in this work is a single classification problem and the loss function applied was a binary cross-entropy (Equation (2)):

$$Loss = -\frac{1}{N}\sum_{i=1}^{N} y_i * log y_i' + (1 - y_i) * log\left(1 - y_i'\right) \tag{2}$$

where $y_i'$ is the i-th scalar value in the model output, $y_i$ is the corresponding target value, and **N** is the number of scalar values in the model output.

A deep learning model is highly dependent on hyperparameters, and hyperparameter optimisation is essential to reach good results. In this work, a CNN based on U-net architecture was built, capable of segmenting volcanic plumes from visible cameras. The values assigned to model parameters are shown in Table 2.

**Table 2.** Hyperparameters required for the training phase for both CNN architectures.

| Hyperparameters Required for Training | |
|---|---|
| **Learning Rate** | 0.0001 |
| **Batch_Size** | 4 |
| **Compile networks** | |
| **Optimiser** | adam |
| **Loss** | binary_crossentropy |
| **Metrics** | Accuracy; iou_score |
| **Fit Generator** | |
| **Step_per_epoch** | 112 |
| **Validation_steps** | 28 |
| **epochs** | 100 |

The encoder and encoder networks contain five layers with the configuration shown in Table 3.

**Table 3.** Convolutional layers description for U-Net architecture.

| Input Layer | | A 2D Image with Shape (768, 768, 3) | | | | | |
|---|---|---|---|---|---|---|---|
| **Encoder Network** | | | | | | | |
| **Convolutional Layer** | **Filters** | **Kernel Size** | **Pooling Layer** | **Activations** | **Kernel Initialiser** | **Stride** | **Dropout** |
| Conv1 | 16 | $3 \times 3$ | yes | ReLU | he_normal | $1 \times 1$ | No |
| Conv2 | 32 | $3 \times 3$ | yes | ReLU | he_normal | $1 \times 1$ | No |
| Conv3 | 64 | $3 \times 3$ | yes | ReLU | he_normal | $1 \times 1$ | No |
| Conv4 | 128 | $3 \times 3$ | yes | ReLU | he_normal | $1 \times 1$ | No |
| Conv5 | 256 | $3 \times 3$ | yes | ReLU | he_normal | $1 \times 1$ | No |
| **Bottle neck** | 512 | $3 \times 3$ | No | ReLU | he_normal | | 0.5 |
| **Decoder Network** | | | | | | | |
| **Convolutional Layer** | **Filters** | **Kernel Size** | **Concatenate Layer** | **Up-Sampling** | **Activations** | **Kernel Initializer** | **Stride** |
| Conv6 | 256 | $3 \times 3$ | Conv5-Conv6 | yes | ReLU | he_normal | $1 \times 1$ |
| Conv7 | 128 | $3 \times 3$ | Conv4-Conv7 | yes | ReLU | he_normal | $1 \times 1$ |
| Conv8 | 64 | $3 \times 3$ | Conv3-Conv8 | yes | ReLU | he_normal | $1 \times 1$ |
| Conv9 | 32 | $3 \times 3$ | Conv2-Conv9 | yes | ReLU | he_normal | $1 \times 1$ |
| Conv10 | 16 | $3 \times 3$ | Conv1-Conv10 | yes | ReLU | he_normal | $1 \times 1$ |
| **Output layer** | 1 | $1 \times 1$ | No | No | Sigmoid | he_normal | |
| **Total trainable params** | | 7.775.877 | | | | | |

The encoder and encoder networks contain five layers with the configuration shown in Table 4.

**Table 4.** Convolutional layers description for SegNet architecture.

| Input Layer | | A 2D Image with Shape (768, 768, 3) | | | | |
|---|---|---|---|---|---|---|
| **Encoder Network** | | | | | | |
| **Convolutional Layer** | **Filters** | **Kernel Size** | **Pooling Layer** | **Activations** | **Stride** | **Dropout** |
| Conv1 | 16 | $3 \times 3$ | yes | ReLU | $1 \times 1$ | No |
| Conv2 | 32 | $3 \times 3$ | yes | ReLU | $1 \times 1$ | No |
| Conv3 | 64 | $3 \times 3$ | yes | ReLU | $1 \times 1$ | No |
| Conv4 | 128 | $3 \times 3$ | yes | ReLU | $1 \times 1$ | 0.5 |
| Conv5 | 256 | $3 \times 3$ | yes | ReLU | $1 \times 1$ | 0.5 |
| **Bottle neck** | 512 | $3 \times 3$ | No | ReLU | | 0.5 |
| **Decoder Network** | | | | | | |
| **Convolutional Layer** | **Filters** | **Kernel Size** | **Up-Sampling** | **Activations** | **Stride** | **Dropout** |
| Conv6 | 256 | $3 \times 3$ | yes | ReLU | $1 \times 1$ | No |
| Conv7 | 128 | $3 \times 3$ | yes | ReLU | $1 \times 1$ | No |
| Conv8 | 64 | $3 \times 3$ | yes | ReLU | $1 \times 1$ | No |
| Conv9 | 32 | $3 \times 3$ | yes | ReLU | $1 \times 1$ | No |
| Conv10 | 16 | $3 \times 3$ | yes | ReLU | $1 \times 1$ | No |
| **Output layer** | 1 | $1 \times 1$ | No | Sigmoid | | No |
| **Total trainable params** | | 11.005.841 | | | | |

In order to show the models built and the difference in the architecture used in this work, Keras provides a function to create a plot of the neural network graph that can make more complex models easier to understand, as is shown in Figure 8.

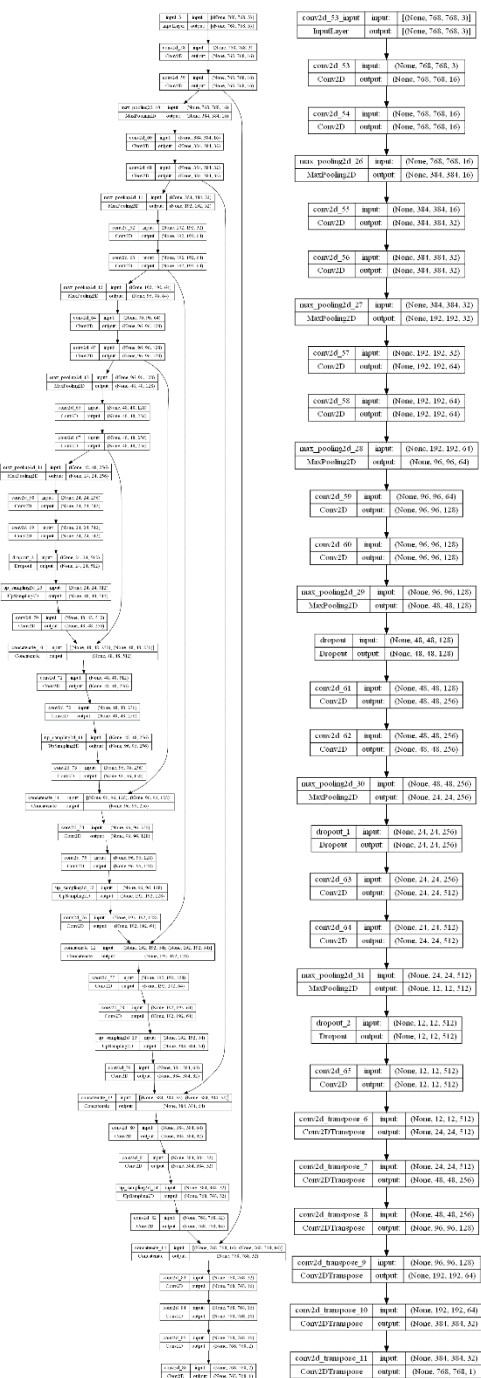

**Figure 8.** Left sketch of the U-net model with Deepest 4, right sketch of the SegNet model (the images are available with higher resolution at the links in [44,45]).

### 4.3. Evaluation of the Proposed Model

Various evaluation metrics are used to calculate the performance of the model. The evaluation metrics used in this research are explained below:

**Accuracy score:** it is the ratio of number of correct pixel predictions to the total number of input samples (Equation (3)).

$$Accuracy = TP/TNP \tag{3}$$

where **TP** is the number of true positives and **NPT** is the total number of predictions.

**Jaccard index** is the Intersection over Union (Equation (4)), where the perfect intersection has a minimum value equal to zero.

$$L(A, B) = 1 - (A \cap B / A \cup B) \qquad (4)$$

where: $(A \cap B / A \cup B)$ is the predicted masks overlap coefficient with the real masks between the union of that masks.

**Validation curves:** the trend of a learning curve can be used to evaluate the behaviour of a model and, in turn, it suggests the type of configuration changes that may be made to improve learning performance [46]. On these curve plots, both the training error (blue line) and the validation error (orange line) of the model are shown. By visually analysing both of these errors, it is possible to diagnose if the model is suffering from high bias or high variance. There are three common trends in learning curves: underfitting (high bias, low variance), overfitting (low bias, high variance) and best fitting (Figure 9).

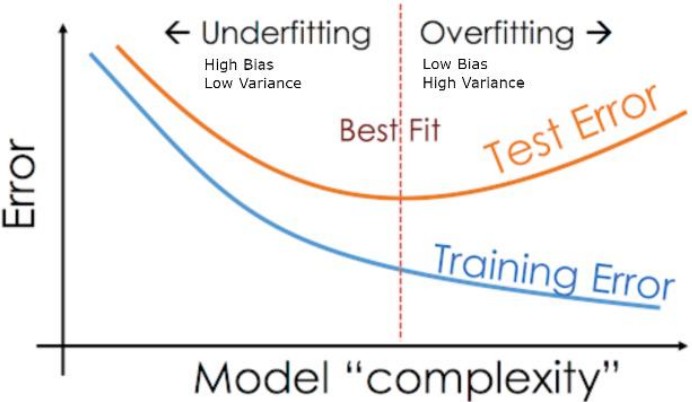

**Figure 9.** Underfitting, overfitting, and best fit example.

Figure 10 shows a trend graph of the cross-entropy loss of both architectures (Y axis) over number of epochs (X axis) for the training (blue) and validation (orange) datasets. For the U-Net architecture, the plot shows that the training process of our model converges well and that the plot of training loss decreases to a point of stability. Moreover, the plot of validation loss decreases to a point of stability and has a small gap with the training loss. On the other hand, for the SegNet architecture, the plot shows that the training process of our model converged well until epoch 30, then showed an increase in variance, taking to a possible overfitting. This means that the model pays a lot of attention to training data and does not generalise on the data that it has not seen before. As a result, the SegNet model performs very well on training data but has more error rates than U-net model on test data.

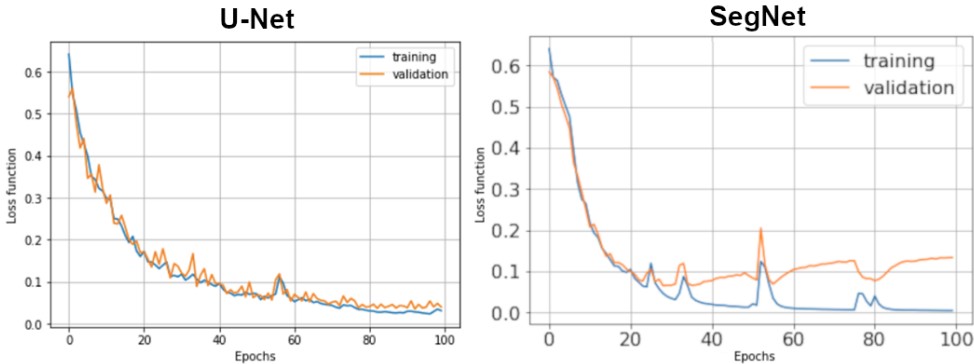

**Figure 10.** Trend curve of loss function.

The loss function for U-Net architecture for the training dataset is 0.026 and validation 0.316 and, for SegNet, for the training dataset is 0.018, while for the validation dataset is 0.142.

Figure 11 shows a trend graph of the accuracy metric (Y axis) over the number of epochs (X axis) for the training (blue) and validation (orange) datasets. In the Epoch 100, the accuracy value reached for the U-Net architecture training dataset is 98.35% and validation dataset is 98.28; while, for SegNet, the accuracy value for the training dataset is 98.15% and validation dataset is 97.56.

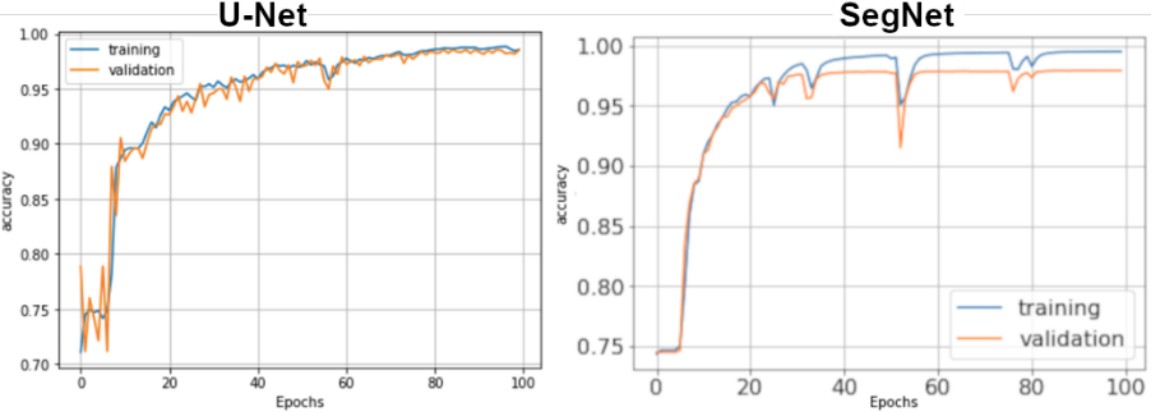

**Figure 11.** Trend curve of accuracy metric of training and validation dataset.

IoU (Intersection over Union) or Jaccard index is the most commonly used metric to evaluate models of semantic segmentation. It is a straightforward metric but extremely effective (metric ranges from 0 to 1, where 1 is the perfect IoU). Thus, in order to quantify the results, for both architectures, the IoUs were calculated using the validation dataset with 112 images with a step of 28 per epoch that represent 20% of the whole dataset. An average of IoU of 0.9013 was obtained for U-Net architecture and, for SegNet, an average value of IoU of 0.88 (Figure 12).

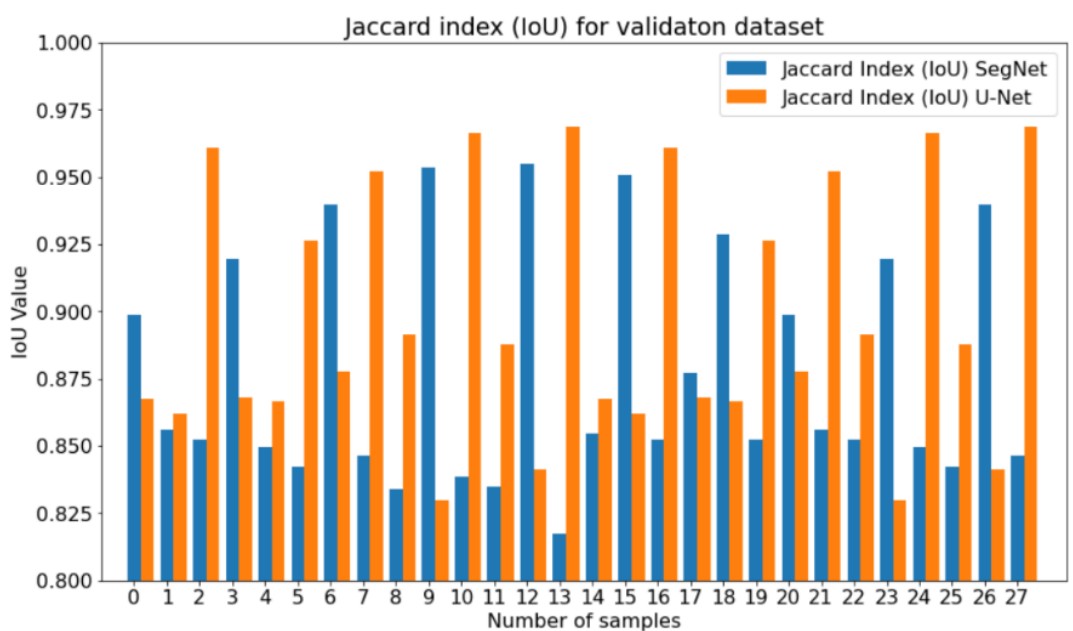

**Figure 12.** Jaccard index percentage for validation dataset of Unet (orange colour) and SegNet (blue colour) architectures.

In Figure 13, the predicted mask results of three samples of the validation dataset are shown, where (a) is the image, (b) is the ground truth mask (mask made by hand), (c) is the predicted mask by SegNet model, and (d) is the predicted mask by U-Net model.

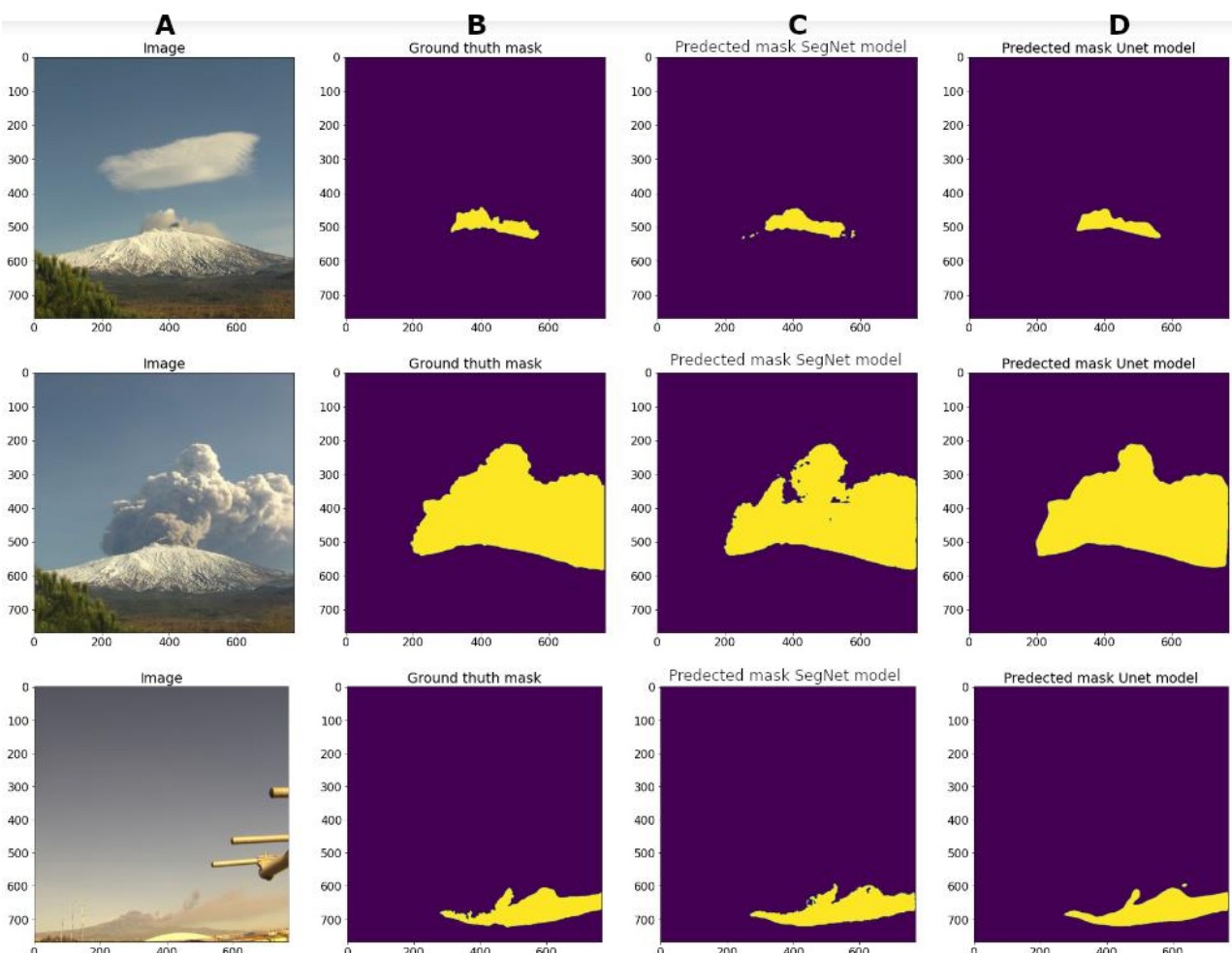

**Figure 13.** Original image (**A**), ground truth mask (**B**), predicted mask by SegNet (**C**), predicted mask by U-net (**D**).

Once the model was completely trained and after verifying training and validation metrics, in order to evaluate how the models performed, a test dataset (data not previously used in training and validation) was used. The samples of the data used provide an unbiased evaluation as the test dataset is the crucial standard to evaluate the model, it is well curated, and it contains carefully sampled data that cover several classes that the trained model will deal with when used in the real world, for example, images non acquired from Etna_NETVIS Network, eruptions in cloudy time, and images from other volcanoes different from Mt. Etna.

Figure 14 shows examples of photographs of different eruptive events, of which two were taken by local citizens during the Etna eruption; the one following belongs to photos of the Monte Cagliato Etna station, the fourth shows the summit crater on a cloudy day, and a last one photo was taken by local people during an eruptive event of the Galeras volcano in Colombia, where the column reached 6 km in height.

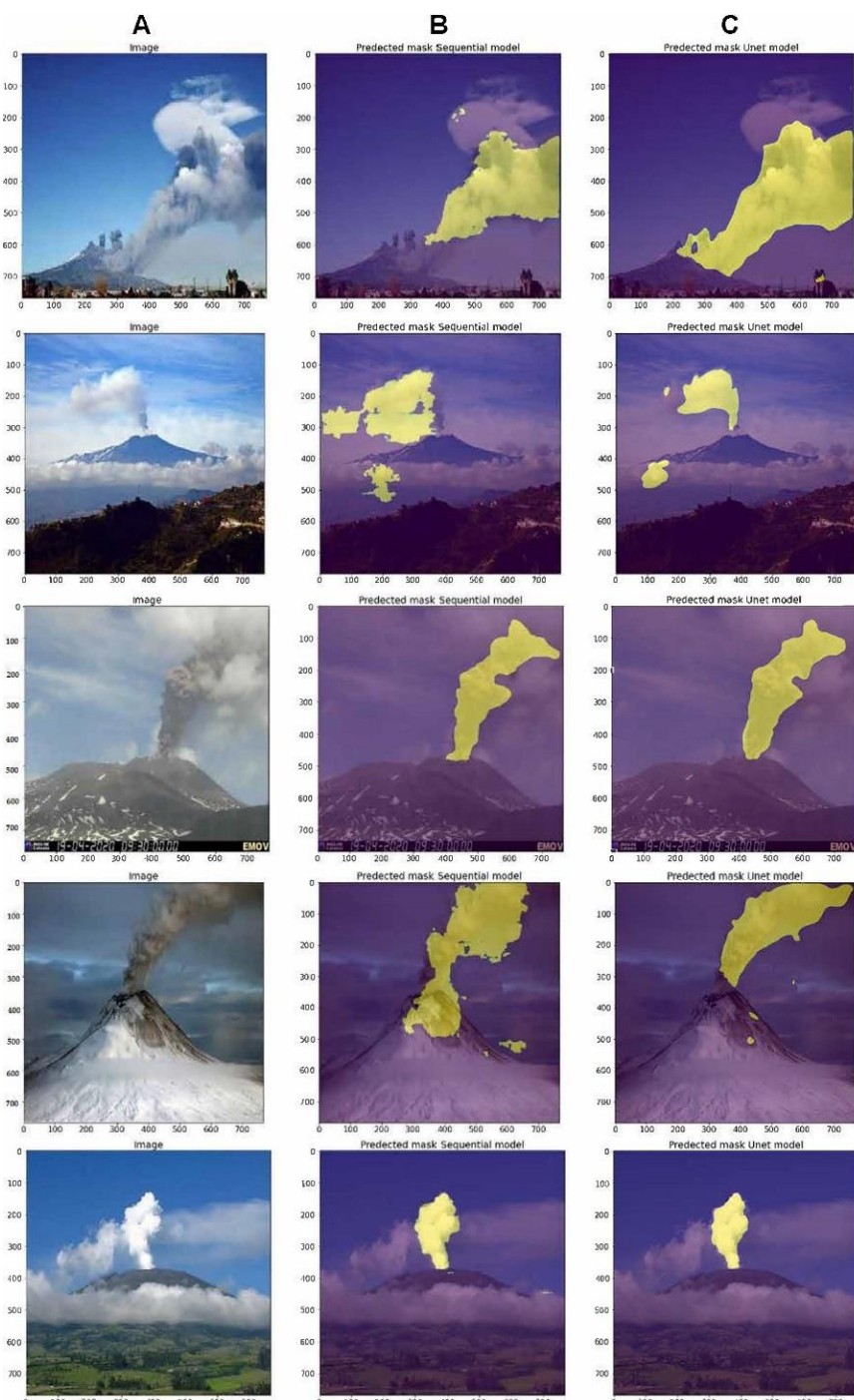

**Figure 14.** Semantic segmentation of results from test dataset: original image (**A**), predicted mask by SegNet (**B**), predicted mask by U-net (**C**).

## 5. Discussion and Concluding Remarks

In this paper, we proposed a new innovative approach based on AI for volcanic monitoring focused on the use of visible high-resolution images coming from a surveillance network of Mount Etna (Etna NETVIS). Considering that optical RGB channels and the wavelength of in situ images carry enough information, the primary aim was using all these data to solve problems related to the characterisation and monitoring of ash plumes during an explosive eruption. For this, a deep convolutional neural network was built to extract ash plume shapes automatically.

Before reaching the final results, we had to face several challenges, as the amount of data was limited; in fact, the accuracy of a neural network largely depends on the quality, quantity, and contextual meaning of training data. Even though our amount of data was limited (560 images), not enough for a model of machine learning, we hypothesised that there could have been a possible overfitting; therefore, to avoid this problem, we artificially increased the amount of data by generating new ones from the existing dataset through "data-augmentation" technique. The use of supervised learning paradigm applied in this work required that the data collected were labelled, and these preprocessing and data labelling tasks were other challenges faced in this work, which took 60% of the whole time of the full project.

In order to assess the performance of our trained deep CNN models, firstly, we measured our model error through metrics combination in a learning curve (training loss and validation loss over time). The training loss indicates how well the model is fitting the training data, while the validation loss indicates how well the model fits new data. Loss measured in the U-Net model error was of 0.026 for the training dataset and 0.0316 for the validation dataset. Secondly, we measured in the learning curve with an accuracy of 0.9835 for the training dataset and 98.28 for the validation dataset, evidencing that our model performance increased over time, which means that the model improved with experience. To reach the optimal fitting during our training, a regularisation named "early stopping" was applied to block our training when detecting an increase in the loss function value, thus avoiding the overfitting. To determine the robustness of our preliminary results, we computed the Jaccard similarity coefficient [47] to measure the similarity and diversity of sample sets. The average (IoU) value obtained from 20% of our validation dataset was equal to 91.3% of similarity. On the other hand, loss measured in SegNet model error was of 0.018 for the training dataset and 0.142 for the validation dataset. In the learning curve, an accuracy of 0.9815 was reached for the training dataset and 97.56 for the validation dataset. These results are interpreted as an increasing model performance over time but giving greater importance to the training data, which means an increase in the value of the variance, leading to possible errors in the segmentation of new data. It should be noted that the SegNet model obtained good results but always lower than those of the U-Net architecture.

The developed method is currently tested for analysis of visible images. As a future work, this method can also be integrated with images acquired from satellite sensors when the terrestrial cameras are out of coverage range. Extensive testing will be performed by exploiting the data of the open-source and on-demand platforms to validate their suitability for different types of explosive volcanoes. Moreover, this is a semi-automatic tool because the data need to be downloaded from a server storage and loaded into the deep NN. Concerning this, the creation of an internal software into the cameras is planned, which can collect and automatically analyse them by deep CNN; this will improve the performance by allowing real-time monitoring and having at disposal a powerful tool in times of emergency.

Predictably, deep learning will become one of the most transformative technologies for volcano monitoring applications. We found that deep CNN architecture was useful for the identification and classification of ash plumes by using visible images. Further studies should concentrate on the effectiveness of deep CNN architectures with large high-quality datasets obtained from remote sensing monitoring networks [25,48].

Concerning the aim of the research in the current phase, the method has been, so far, developed for plume monitoring purposes, such as detection and measurement of ash clouds emitted by large explosive eruptions, focusing on the capability of measuring the height of the plume, as the most relevant parameter to understand the magnitude of the explosion, and not yet for observing eruption precursors. By extending the procedure to process large time series of images, additional parameters can be extracted, such as elevation increase rate and temporal evolution, which can significantly contribute to set up a low-cost monitoring tool to help mitigate volcanic hazards. Furthermore, additional precious information usable as precursor indices can be derived from the monitoring of the

degassing state of volcanoes. As is already noticeable in Figure 14, the algorithm allowed the distinction of a lenticular meteorological cloud from volcanic water vapor emission, excluding it from the eruption ash plume. These water vapour clouds can give important indications about changes in a volcano's degassing, considered as eruption precursors, so their discerning may be profitable for the mitigation of risks in volcanic context. However, the data used in this research are still insufficient and inadequate to detect other parameters as indicators of dew point or humidity. The important difference is that a large eruption plume is recognizable from the meteorological clouds in the background. Conversely, the degassing plume is subject to the physical condition of the atmosphere.

The results shown in this work demonstrated that this innovative approach based on deep learning is capable of detecting and segmenting volcanic ash plume and can be a powerful tool for volcano monitoring; also, the proposed method can be widely used by volcano observatories, since the trained model can be installed on standard computers where they can analyse images acquired by either own surveillance cams or from other sources through internet, as long as visibility allows, enhancing the observatory capacity in volcano monitoring.

**Author Contributions:** J.F.G.T. developed the neural network and performed the analysis in this chapter under the supervision of M.M. and M.C. as principal tutors; J.F.G.T. prepared the original draft; J.A.P.B., A.C., M.M. and M.C. contributed to the writing, review, and editing of the manuscript. All authors have read and agreed to the published version of the manuscript.

**Funding:** This research was conducted during a PhD course, with a studentship by CEIBA Colombia foundation (https://ceiba.org.co/ (accessed on 1 August 2022)), the APC was funded by Istituto Nazionale di Geofisica e Vulcanologia (INGV).

**Data Availability Statement:** Etna eruption 24-12-2018 dataset is curated by INGV Osservatorio Etneo Catania and is available on request (https://www.ingv.it (accessed on 1 August 2022)). Requests to access these datasets should be directed to https://www.ingv (accessed on 1 August 2022). Data presented in this study are available upon request from the corresponding author. The data is not publicly available due to source for security policy is not possible to access to data from external.

**Acknowledgments:** Dataset was obtained from INGV; The neural network was training in laboratory of Department of Civil, Building and Environmental Engineering of Sapienza of Roma university. We thank INGV for financial support for publishing this paper. We thank reviewers for their comments on an earlier version of the paper.

**Conflicts of Interest:** The authors declare no conflict of interest.

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
