# Peer review of "Convolutional Neural Network Algorithms for Semantic Segmentation of Volcanic Ash Plumes Using Visible Camera Imagery"

_remotesensing, doi:10.3390/rs14184477_

Round 1

Reviewer 1 Report

This is a great paper and a step forward in monitoring.  However, I would point that out in the paper.  I'd be specific on a few things:

(1) This can be automated for use in volcano monitoring, particularly in places where more expensive instrumentation is not available, such as seismometers or infrasound or IR cameras.

(2) This can go beyond simple eruption detection.  Precursors as well as eruption dynamics could be gleaned from this data.

To build on this last point, I notice that the water vapor clouds are excluded from the plume in the algorithm.  For example, the lenticular cloud in figure 14.  This is good, however changes in water vapor clouds can also be used (in concert with dew point and humidity) to be a proxy for changes in a volcano's degassing to look for eruption precursors.  (even sometimes the sudden absence of a persistent steam plume)

At night, glow could be detected in webcam imagery.

But most importantly, the height of the plume could be determined from the imagery, knowing the camera's distance from the source and the characteristics of its lens.  In addition to height, rise rate, eruption time and duration could also be estimated by looking at time sequential images.  This is great information for mitigation and to be used in plume tracking models.

I don't suggest the authors do all that for this paper by any means.  However, they should mention the applications of this technique to provide a low-cost monitoring tool to help mitigate volcanic hazards.

So adding a sentence here and there and a paragraph at the end to address this and the paper would be near perfect.

I've attached an annotated pdf.

Author Response

Response to Reviewer 1 Comments

Point 1: I would point out in the paper that the approach can be automated for use in volcano monitoring, particularly in places where more expensive instrumentation is not available, such as seismometers or infrasound or IR cameras

Point 2: I would point out in the paper that the approach can go beyond simple eruption detection. Precursors as well as eruption dynamics could be gleaned from this data.

To build on this last point, I notice that the water vapor clouds are excluded from the plume in the algorithm for example, the lenticular cloud in figure 14. This is good, however changes in water vapor clouds can also be used (in concert with dew point and humidity) to be a proxy for changes in a volcano's degassing to look for eruption precursors (even sometimes the sudden absence of a persistent steam plume). At night, glow could be detected in webcam imagery.

Response: The method has been so far developed for plume monitoring purposes, such as detection and measurement of ash clouds emitted by large explosive eruption, and not yet for observing precursors such as change in degassing state. The important difference is that large eruption plume is recognizable from the meteorological clouds in the background. Conversely, the degassing plume is subject to the physical condition of the atmosphere. A completely new study could be done to compare degassing plumes in different meteorological conditions (temperature, wind velocity, dew point and humidity) and use it to estimate important variations in volcanic gas rate.

But most importantly, the height of the plume could be determined from the imagery, knowing the camera's distance from the source and the characteristics of its lens. In addition to height, rise rate, eruption time and duration could also be estimated by looking at time sequential images. This is great information for mitigation and to be used in plume tracking models. I don't suggest the authors do all that for this paper by any means. However, they should mention the applications of this technique to provide a low-cost monitoring tool to help mitigate volcanic hazards.

Response: In this work, we focused on the capability of measuring the height of the plume, which is the most relevant parameter to understand the magnitude of the explosion. By extending the procedure to process large time series of images, additional parameters such as elevation increase rate and temporal evolution can significantly contribute to set-up a low-cost monitoring tool to help mitigate volcanic hazards.

We thank the reviewer for the constructive comments. We followed the suggestions by adding content on these aspects.

Several statements have been added or reformulated in section “5. Discussion and concluding remarks” to remark these points underlined by the reviewer #1:

  • Pages 22-23, lines479-498 : Concerning the aim […] gas rate.

Reviewer 2 Report

Interesting paper that design a path to apply neural networks to early warnings of volcanic ash in atmosphere. 

The aim of the paper is to asses if it is possible to use the monitoring cameras that are already in place to monitor volcanic activity to detect volcanic ash emission. 

The possible use will be an automated volcanic ash emission alert.

The paper uses images of an already operating network to develop an algorithm. 

Already well known techniques are used to treat the images to increase their number to for the training and to analyse the goodness of the results.

Results are positive and demonstrate that the procedure can be useful if applied properly to monitor volcanic ash emission, probably, in particular, in case of volcanoes that are isolated. 

The paper well describes the aim, procedure and results. 

A key point in research like this is the sample preparation and use.  

Authors in this case are helped from the fact that the camera sites are fixed, that means that the landscape can be easily defined in the training images. Nevertheless the example images showed in the paper appears to be in clear sky condition so very simple in terms of “detection”. 

My question is: do the authors made a special analysis of the quality of results in the case of presence of clouds? Can the author give a figure and compare with the favourable case of clear sky? 

I would add also those information to the paper.

After that i consider the paper deserve publication on remote sensing.

Author Response

Response to Reviewer 2 Comments

Point 1: do the authors made a special analysis of the quality of results in the case of presence of clouds? Can the author give a figure and compare with the favourable case of clear sky?

 Response 1: Some tests have been done to evaluate the performance of the trained model on images with presence of clouds and showed a higher percentage of pixels of misdetection, especially as False Positives but showing high coherence on True Positives. To improve the results of this first work is necessary to increase the training dataset using images in the presence of meteorological clouds.

Two statements have been added on paragraph “ 4.3. Evaluation of the proposed model” to discuss this point underlined by the reviewer #2.

  • Page 17, lines 409-416: Once the model […] from Mt. Etna.

Furthermore, one picture of the same paragraph has been modified, also showing two examples with clouds.

  • Page 18, Figure 17.

Reviewer 3 Report

You spend a lot of text on Etna, that can be shortened by half, This paper is methotology and you mis-x later in Sromboly and Galeras, 

The aim is good to detect volcanic plume, but it would be interesting to see if the analysis can be done in more than 2 dimensions?

surely your method can detect volcanic plume on a image and as such this first step is good for future research in the field.

Author Response

Response to Reviewer 3 Comments

Point 1: You spend a lot of text on Etna, that can be shortened by half, This paper is methotology and you mis-x later in Sromboly and Galeras

Response 1: The text relating to the geology of Etna in paragraph 2 has been shortened as suggested. The application of the methodology to other case studies, such as that of Galeras and Stromboli, has been deliberately reported to show the validation of the method on other study cases.

Several statements have been cut or reformulated in section “2. Geological settings”:

  • Page 3, lines 82-108: Mt. Etna […] emergency planning.

Point 2: The aim is good to detect volcanic plume, but it would be interesting to see if the analysis can be done in more than 2 dimensions? Surely your method can detect volcanic plume on a image and as such this first step is good for future research in the field.

Response 2: The work focuses on the use of surveillance cameras that do not allow 3D model extraction due to poor overlap, unfavourable baseline, and low image resolution. Simulation of the camera network geometric and sensor configuration have been carried out in MEDSUV project and will be adopted as a reference for future implementation of the Etna Network.

Two statements and one reference to bibliography have been added on page 4, paragraph “3. Etna_NETVIS Network” to discuss this point underlined by the reviewer #3.

  • Page 4, lines 130-135: These surveillance cameras […] of Etna Network.

Reviewer 4 Report

The manuscript entitled "Convolutional Neural Networks Algorithms for Semantic Segmentation of Volcanic Ash Plumes Using Visible Camera Imagery", presents an innovative proposal of the use of binary semantic segmentation applied to identification of volcanic ash plumes from images of a camera that records the visible spectrum.

The innovative part is limited exclusively to the scope of application of semantic segmentation, otherwise the article is missing:

* greater state of the art identifying other geophysics, or earth observation scenarios where these semantic segmentation techniques are used.

* More state of the art semantic segmentation models more novel and current than the U-Net architecture. They are more modern and there are more models to experiment with, compare and thus justify the choice of a Deep Learning model or architecture for this purpose.

* Larger data set. The number of labeled images are few, even using the data augmentation technique.

* It is also not clear what is intended to be identified in the camera images: to identify a type of eruption, e.g. stronbolian, to identify any type of eruption?

There are also leftovers:

* as the article does not have the focus on neural networks and Deep Learning there is no need to devote several pages to explain the multilayer perceptron or the architecture of the U-Net network. These concepts are perfectly described in other articles and texts and can be referenced.

Honestly, the research behind the manuscript requires more effort and time to do so. More than the usual time provided to do a major review. For this reason I propose to reject and resubmit when you have addressed the issues listed.

Author Response

Response to Reviewer 4 Comments

Point 1: the article is missing greater state of the art identifying other geophysics, or earth observation scenarios where these semantic segmentation techniques are used.

Response 1: The review of state of the art performed for the setting of this work focused on the studies that best fit the topic of study. Despite this, several refererences have been added at pag. 2 of the section 1. Introduction, to extend the review of the state of the art to other areas of environmental monitoring, following the suggestion of the reviewer 4.

  • Page 2, lines 40: addition of reference 9.
  • Page 2, lines 45: addition of reference 10.
  • Page 2, lines 47-53: Addition from reference 11-14 and addition of statement supporting the state of art.

Point 2: the article is missing more state of the art semantic segmentation models more novel and current than the U-Net architecture. They are more modern and there are more models to experiment with, compare and thus justify the choice of a Deep Learning model or architecture for this purpose.

Response 2: The approach used is the most suitable for this phase of work; for future development, we propose to apply a more advanced technique. In fact, we preferred to use a consolidated technique, this being the first approach. We will then move on to more innovative and, therefore, more experimental methods. Nevertheless three references have been added pag. 2 of the section 1., in order to consider other publications in the volcanological field, following this comment, which however make use of minor architecture.

  • Page 2, lines 55-59: addition of reference 15-22.
  • Page 10, lines 282: addition of reference 32-35.

Point 3: the article is missing larger data set. The number of labeled images are few, even using the data augmentation technique.

Response 3: The analysed dataset was not collected specifically for applying any automatic detection method. The work mainly aims to understand the feasibility and the problems/critical issues for developing an operational monitoring system to be included in the standard surveillance activity of a volcano observatory, such as INGV-EO.

Point 4: It is also not clear what is intended to be identified in the camera images: to identify a type of eruption, e.g. stronbolian, to identify any type of eruption?

Response 4: The objective is mainly to obtain quantitative measures (height and width, rate of expansion ….) of the plumes produced by large explosive eruptions like lava fountaining of Mt. Etna. The understanding of the eruption styles is out of the scope of the paper. For instance, strombolian activity produces small explosions whose dispersion has a significant impact neither on the air traffic around the volcano nor on the vehicular traffic on its flanks, and it is confined to the areas around volcanic vents that are not occupied by anthropogenic activities but only by occasional visitors.

Point 5: as the article does not have the focus on neural networks and Deep Learning there is no need to devote several pages to explain the multilayer perceptron or the architecture of the U-Net network. These concepts are perfectly described in other articles and texts and can be referenced.

Response 5: since the work is aimed at applying A.I. to volcanic monitoring, we considered an exhaustive explanation of the method important, also addressed to operators in the sector.

Round 2

Reviewer 4 Report

The authors have replied to the comments of the first review, including some more references to the state of the art and justifying why they have not taken into account the comments of the first review.  These justifications are not convincing. If the manuscript had been submitted to a proper journal of volcanology or geophysics it might make sense to include the perceptron descriptions or all the description related to the downward gradient optimization or the description of the convolutional network. But in this journal there are numerous articles related to semantic segmentation applied to different types of images (not only visible and static cameras) and it is not considered necessary for the reader to include all this information that does not contribute more than state of the art. On the other hand, I do consider that it is more important for the reader to see justified the choice of the selected network/architecture within the currently existing ones to investigate or answer questions in the field that concerns them. To this proposal the authors have avoided and postponed to the following research / publications to make a comparison. It is not indicated at any time which technology or libraries have been used to carry out the experiments. It is not indicated how many epochs the networks have been trained, statistical results of repeating the experiments several times are not shown in order to know how the trainings converge and to be able to keep the best of the results. In short, the authors do not justify by means of results that this proposal is the one that could be used in a practical way for the type of problem they are trying to help monitor.

Author Response

1. Reviewer #4 - "The authors replied to the comments of the first review including some more references to the state of the art and justifying why they have not taken into account the comments of the first review.  These justifications are not convincing.”.

In the current version of the manuscript, the authors have worked to strenghten the descriptions of the applied method. In particular, to discuss other models in addition to the adopted U-Net architecture, in the current version the results obtained for videocamera images of Monte Etna using the SegNet architecture are shown. The comparison shows that the U-Net method provide satisfactory results demonstrating to be adequate to the needs of the present surveillance network. However, future development will certainly let to a refinment of the segmentation technique trough the comparison of multiple architectures.

In agreement with this modification, the paper has been edited as follows:

  • The utilization of two ANN architectures has been indicated in the Abstract (“Two well-established architectures, the segNet and the U-Net, have been used for the processing of in-situ images, to validate their usability in the field of volcanology“).
  • A brief introduction to the efficience of the U-Net architecture has been added in section “1. Introduction”, pag. 2 (lines 47-51): “Specifically, […] FCN type [ , ]”.
  • The comparison of both methods has also been introduced in the same section, pag. 2 (lines 75-79): “[…] the training of two very consolidated […] volcanological field.”.
  • At the end of the section “1. Introduction”, pag. 3 (lines 84-88), authors highlight the possible potential of future research on the creation of new networks for classification in line with the comments of Reviewer #1: “The current results […] the creation of more advanced classification networks […] change in degassing state.”.
  • Both architectures, SegNET and U-Net, have been referenced in section “4.2.1. Convolutional Neural Network Architectures”, pag. 8 (lines 213-215): “Different algorithms […] segNet [] and U-Net [].”.
  • U-Net has been summarized in section “4.2.1. Convolutional Neural Network Arcitectures”, pags. 8-9, (lines 223-232): “The U-Net […] possible to accept images of any size.”.
  • SegNet has been summarized in section “4.2.1. Convolutional Neural Network Architectures”, pag. 9, (lines 238-242): “On the other hand, […] layer in the decoder.”.
  • The architectures of U-Net and SegNet are presented in Tables 3 and 4, respectively.
  • The sketch of the SegNet model has been added to Figure 8.
  • The trend curve of loss function for the SegNet architecture has been included in Figure 10, and introduced in the reformulated lines 323-329, pag. 13: “On the other hand, for the SegNet […] shown in Figure 13.”.
  • The trend curve of accuracy metric for SegNet architecture has been included in Figure 11, introduced at line 342-344, page 14: “[…] while for SegNet […] is 97.56.”.
  • The Jaccard index percentage for SegNet has been included in Figure 12, introduced at lines 355, page 14: “[…] SegNet an average value of IoU of 0.88 (Figure 12).”.
  • The predicted mask by SegNet has been included in Figure 13, introduced at lines 363, pag 15: “[…] (c) is the predicted mask by SegNet model […]”.

A discussion of the result comparison has been added to section “5. Discussion and concluding remarks”, page 20 (lines 419-426): “By the other hand, Loss measured in SegNet […] lower than those of the U-Net architecture.”.

2. Reviewer #4 - "If the manuscript had been submitted to a proper journal of volcanology or geophysics it might make sense to include the perceptron descriptions or all the description related to the downward gradient optimization or the description of the convolutional network. But in this journal, there are numerous articles related to semantic segmentation applied to different types of images (not only visible and static cameras) and it is not considered necessary for the reader to include all this information that does not contribute more than state of the art.”

Following this comment the text was modified by eliminating unnecessary contents. In particular, sections “4.2. Methods: ANN and UNET”, was edited citing and referencing common concepts for this journal.

3. Reviewer #4 - “On the other hand, I do consider that it is more important for the reader to see justified the choice of the selected network/architecture within the currently existing ones to investigate or answer questions in the field that concerns them. To this proposal the authors have avoided and postponed to the following research / publications to make a comparison. It is not indicated at any time which technology or libraries have been used to carry out the experiments.”

 To provide this justification to the reader, authors have done the modifications already indicated in Response 1.

4. Reviewer #4 - “It is not indicated how many epochs the networks have been trained, statistical results of repeating the experiments several times are not shown in order to know how the trainings converge and to be able to keep the best of the results. "

 Authors recognize the importance of statistics measurements to verify the quality of the results. Accordingly, in Figure 10 and 11 the number of epochs with the loss function and accuracy metrics, were added. Moreover, authors has described the evolution and convergence of the training phase in lines 319-329, pag. 13: “For the U-Net architecture, […] as shown in Figure 13.”. Similarly, the evolution of the accuracy with the number of epochs has been descrived in lines 339-344, pag. 14: “Figure 11 shows a trend […] and validation is 97.56].”. In addition, graphs and descriptions are referred to both architectures, SegNet and U-Net, providing data comparison.

5. Reviewer #4 - “In short, the authors do not justify by means of results that this proposal is the one that could be used in a practical way for the type of problem they are trying to help monitor.”

Considering that a lot of volcanoes are observed using standard (visible band) video/web cams operated by both Volcano Observatory and private entities for several scopes (tourism, traveling, entertainment, etc.), we developed a tool based on Artificial intelligence for detecting volcanic plumes produced by large eruptions that is tailored for these widely adopted cams. This tool allows to detect the plume and to calculate automatically geometric parameters that are very useful to understand the magnitude of an ongoing eruption. Statistical data reported in this work justify the capability of the developed tool as described in Response 4.

On the other hand, the proposed method can be widely used by Volcano Observatories having low budgets, since the trained model can be installed on standard computers where they can analyze images acquired by either their own surveillance cams or other sources via internet, as long as the visibility allows it, enhancing the observatory capacity in volcano monitoring.
